# Glycine-Induced Phosphorylation Plays a Pivotal Role in Energy Metabolism in Roots and Amino Acid Metabolism in Leaves of Tea Plant

**DOI:** 10.3390/foods12020334

**Published:** 2023-01-10

**Authors:** Yuchen Li, Kai Fan, Jiazhi Shen, Yu Wang, Anburaj Jeyaraj, Shunkai Hu, Xuan Chen, Zhaotang Ding, Xinghui Li

**Affiliations:** 1College of Horticulture, Nanjing Agricultural University, Nanjing 210095, China; 2Tea Research Institute, Qingdao Agricultural University, Qingdao 266109, China; 3Tea Research Institute, Shandong Academy of Agricultural Sciences, Jinan 250100, China

**Keywords:** tea plants, organic nitrogen, glycine, protein phosphorylation, metabolic pathway

## Abstract

Phosphorylation is the most extensive post-translational modification of proteins and thus regulates plant growth. However, the regulatory mechanism of phosphorylation modification on the growth of tea plants caused by organic nitrogen is still unclear. In order to explore the phosphorylation modification mechanism of tea plants in response to organic nitrogen, we used glycine as the only nitrogen source and determined and analyzed the phosphorylated proteins in tea plants by phosphoproteomic analysis. The results showed that the phosphorylation modification induced by glycine-supply played important roles in the regulation of energy metabolism in tea roots and amino acid metabolism in tea leaves. In roots, glycine-supply induced dephosphorylation of proteins, such as fructose-bisphosphate aldolase cytoplasmic isozyme, glyceraldehyde-3-phosphate dehydrogenase, and phosphoenolpyruvate carboxylase, resulted in increased intensity of glycolysis and decreased intensity of tricarboxylic acid cycle. In leaves, the glycine-supply changed the phosphorylation levels of glycine dehydrogenase, aminomethyltransferase, glutamine synthetase, and ferredoxin-dependent glutamate synthase, which accelerated the decomposition of glycine and enhanced the ability of ammonia assimilation. In addition, glycine-supply could improve the tea quality by increasing the intensity of amino acids, such as theanine and alanine. This research clarified the important regulatory mechanism of amino acid nitrogen on tea plant growth and development through protein phosphorylation.

## 1. Introduction

Tea is the mainstream beverage in the world. Tea plants [*Camellia sinensis* (L.) O. Kuntze] could absorb and utilize organic nitrogen [1], but there are few related studies. Many studies focus on the absorption and utilization of ammonium nitrogen and nitrate nitrogen by tea plants while ignoring that organic nitrogen is also an important nitrogen source for the growth of tea plants. Today, an increasing number of studies have shown that organic nitrogen contributes to plant nitrogen nutrition [2,3], and many terrestrial plants including desert shrubs [4], eucalyptus [5], Wollemi pine [6], *D. longan* Lour. and *E. japonica* Lindl. [7], can absorb intact low-molecular-weight organic nitrogen directly. Amino acid nitrogen is a major fraction of the low-molecular-weight organic nitrogen in the soil. In addition to being a nitrogen source, amino acid nitrogen is also a signal substance regulating plant growth. Among them, glycine (Gly) has become the model compound in many plants’ organic nitrogen uptake studies because of its high content, high water solubility, and low molecular weight [8,9]. Lots of studies have shown that Gly can affect the growth of plant roots and leaves [2,10,11]. Nevertheless, the effect of Gly as a nitrogen source on the growth and development of tea plants is still unclear.

The growth of plants is regulated at various molecular levels, such as transcription, metabolism, and translation. In the past several years, it has been found that the post-translational modification of protein plays an important role in regulating the growth of plants [12,13]. Among them, phosphorylation, the most extensive post-translational modification of proteins, plays an irreplaceable role in regulating the growth of plants. For instance, phosphorylation altered the activity of sucrose synthase and thus regulated the sucrose metabolism of *Vicia faba* [14]. The response of soybean root hair to rhizobial inoculation was through the formation of a complex network of kinase-substrate and phosphatase-substrate interactions [15]. So, we hypothesized that protein phosphorylation is a good entry point to explore the regulation of Gly on the growth of tea plants. Fortunately, phosphoproteomics has developed rapidly in recent years, it aims to accomplish a comprehensive analysis of protein phosphorylation through detecting phosphoproteins and their phosphorylated amino acid residues in a qualitative and quantitative fashion. Phosphopeptide enrichment, combined with high-accuracy mass spectrometry (MS) and related bioinformatics tools, has been used for large-scale phosphoproteomic analyses of various plants [16,17,18,19,20]. In addition, metabolomic analysis is also a powerful and extensively used technology for comprehensive profiling and comparison of metabolites in plant metabolism, and it is widely used on tea plants [21,22]. The changes in metabolites can reflect the regulation results of phosphorylation to a certain extent. Therefore, the combination of phosphoproteomics and metabolomics will be an important means to explore the regulation mechanism of Gly on the growth of tea plants.

In this study, the effects of Gly-supply on tea plant growth were investigated in hydroponic conditions. Firstly, the contents of amino acids were determined to show the nitrogen metabolism in tea plants. Secondly, the photosynthesis intensity and enzyme activity were determined to reflect the physiological condition of the tea plant. Finally, quantitative phosphoproteomics and metabolomics were applied to analyze the phosphorylated proteins and metabolites of tea plants with Gly-supply. This study not only clarified the modification mechanism of protein phosphorylation in the process of glycine absorption by tea plants but also clarified the important regulatory mechanism of amino acid nitrogen on the growth and development of tea plants, establishing an important theoretical foundation for the utilization of amino acid nitrogen by tea plants.

## 2. Materials and Methods

### 2.1. Plant Materials and Treatments

Two-year-old clonal tea seedlings (*Camellia sinensis* cv. ‘Lianshan 1′) were hydroponically cultivated in the greenhouse at the Tea Research Institute, Qingdao Agricultural University in Shandong Province of China (36°19′ N, 120°23′ E). The growth conditions were set as follows: 28 ± 1 °C/20 ± 1 °C (16 h day/8 h night), 280 μM·m^−2^·s^−1^ photon flux densities, 80 ± 5% relative humidity. The nutrient solution contained 1.00 mM (NH4)_2_SO_4_, 0.10 mM KH_2_PO_4_, 0.45 mM K_2_SO_4_, 0.40 mM MgSO_4_, 0.30 mM CaCl_2_, 6.30 μM FeEDTA, 10.00 μM H_3_BO_3_, 1.50 μM MnSO_4_, 1.00 μM ZnSO_4_, 0.20 μM CuSO_4_, and 0.50 μM (NH_4_)_6_Mo_7_O_24_ (Appendix A). At first, the hydroponic seedlings were grown in the normal nutrient solution for 2 weeks; then, the tea seedlings were divided into two groups. One group was transferred to the nitrogen deficiency nutrient solution (removed (NH4)_2_SO_4_) as the control group (CK), and the other group was transferred to the nutrient solution with glycine (1.00 mM) instead of (NH4)_2_SO_4_ as the experimental group (Gly). An appropriate amount of ampicillin sodium (about 0.0125 g/L) was used to inhibit the growth of microorganisms. After 20 days, when new roots appeared in the control group, but almost no new roots were formed in the experimental group, the third mature leaves from the top were harvested as leaf samples (tea leaves of CK labelled as CL, tea leaves with Gly-supply labelled as GL) and the whole roots were harvested as root samples (tea roots of CK labelled as CR, tea roots with Gly-supply labelled as GR). Each sample consisted of three biological replicates. All samples were washed with distilled deionized water, then rapidly frozen in liquid nitrogen and stored at −80 °C for further study.

### 2.2. Determination of Physiological Indices in Leaves and Roots

Physiological indices including photosynthetic rate, chlorophyll contents, and enzyme activities were measured in tea seedlings. The photosynthetic rate was determined by the LI-6400XT portable photosynthesis system and chlorophyll contents by the SPAD-502 Chlorophyll Meter Model. The enzyme activities of glutamine synthetase (GS), ferredoxin-dependent glutamate synthase (Fd-GOGAT), sucrose synthase (SuSy), trehalose-phosphate synthase (TPS), phosphoglucomutase (PGM), hexokinase (HK), fructose-bisphosphate aldolase cytoplasmic isozyme (ALDO), glyceraldehyde-3-phosphate dehydrogenase (GAPDH), phosphoenolpyruvate carboxylase (PEPC), phosphoenolpyruvate carboxykinase (PEPCK) and NADP-dependent malic enzyme (NADP-ME) were determined using kits from Suzhou Grace Biotechnology Co., Ltd. (Suzhou, China). The enzyme activity of SuSy included decomposition activity (SuSy-I) and synthesis activity (SuSy-II).

### 2.3. Determination of Amino Acids Contents and Total Nitrogen Contents

The contents of total nitrogen were performed using the Kjeldahl method according to the State Standard of China and recorded as NY/T 2419-2013. The contents of 19 amino acids were determined by HPLC. Approximately 0.2 g sample was homogenized in 1.5 mL of 6 mol/L hydrochloric acid solution. One microliter of the homogenate was transferred to a hydrolysis tube, filled tube with nitrogen for 5 min, then sealed and hydrolyzed at 110 °C for 18 h. The hydrolyzed sample was transferred to a 1.5 mL centrifuge tube and centrifuged for 5 min at 12,000 rpm (RWD M1324, Shenzhen, China). The supernatant (0.5 mL) was transferred to a new centrifuge tube and dried in a vacuum. Then, 0.5 mL 0.1 mol/L hydrochloric acid solution was added to the tube to dissolve the sample completely. The tube was then centrifuged at 12,000 rpm for 5 min. Two hundred μL of the supernatant was mixed with 20 μL norleucine, 100 μL triethylamine solution, and 100 μL phenyl isothiocyanate (PITC) solution and left at room temperature for 1 h. Finally, 400 μL n-hexane was added and centrifuged at 12,000 rpm for 2 min after oscillation. The injection volume of the HPLC system (Wufeng LC-100) was 10 µL and the column used was a 5 µm Diamonsil AAA Column (250 mm × 4.6 mm) at 35 °C. Mobile phase A was 0.05 mol/L sodium acetate aqueous solution, and mobile phase B was methanol/acetonitrile/water [20:60:20 (*v/v/v*)]. The mobile phase gradient is shown in Appendix A. The flow rate was 1 mL/min, and the eluate was monitored at 254 nm.

### 2.4. Metabonomic Analysis (UPLC-MS/MS)

#### 2.4.1. Sample Preparation and Extraction

The tea samples were freeze-dried using a vacuum freeze-dryer (Scientz-100F) and homogenized to a fine powder by a mixer mill (MM 400, Retsch). A total of 100 mg of lyophilized powder was dissolved in 1.2 mL 70% methanol solution, and vortexed 30 s every 30 min for 6 times. Then, the samples were placed in a refrigerator at 4 °C overnight. Following centrifugation at 12,000 rpm for 10 min, the extracts were filtrated (SCAA-104, 0.22 μm pore size; ANPEL, Shanghai, China, http://www.anpel.com.cn/ accessed on 20 September 2021) before UPLC-MS/MS analysis.

#### 2.4.2. UPLC Conditions

The sample extracts were analyzed using UPLC-ESI-MS/MS system (UPLC, SHIMADZU Nexera X2, https://www.shimadzu.com.cn/, accessed on 20 September 2021); MS, Applied Biosystems 4500 Q TRAP, https://www.thermofisher.cn/cn/zh/home/brands/applied-biosystems.html, accessed on 20 September 2021). The analytical conditions were as follows, UPLC: column, Agilent SB-C18 (1.8 µm, 2.1 mm × 100 mm). The mobile phase consisted of solvent A, pure water with 0.1% formic acid, and solvent B, acetonitrile with 0.1% formic acid. Sample measurements were performed with a gradient program that employed the starting conditions of 95% A and 5% B. Within 9 min, a linear gradient of 5% A and 95% B was programmed, and a composition of 5% A and 95% B was kept for 1 min. Subsequently, a composition of 95% A and 5% B was adjusted within 1.1 min and kept for 2.9 min. The flow velocity was set as 0.35 mL per minute; the column oven was set to 40 °C; the injection volume was 4 μL. The effluent was alternatively connected to an ESI-triple quadrupole-linear ion trap (QTRAP)-MS.

#### 2.4.3. ESI-Q TRAP-MS/MS

LIT and triple quadrupole (QQQ) scans were acquired on a triple quadrupole-linear ion trap mass spectrometer (Q TRAP), AB4500 Q TRAP UPLC/MS/MS System, equipped with an ESI Turbo Ion-Spray interface, operating in positive and negative ion mode and controlled by Analyst 1.6.3 software (AB Sciex). The ESI source operation parameters were as follows: an ion source, turbo spray; 550 °C source temperature; ion spray voltage (IS) 5500 V (positive ion mode)/−4500 V (negative ion mode); ion source gas I (GSI), gas II (GSII) and curtain gas (CUR) were set at 50, 60, and 25.0 psi, respectively; the collision-activated dissociation (CAD) was high. Instrument tuning and mass calibration were performed with 10 and 100 μmol/L polypropylene glycol solutions in QQQ and LIT modes, respectively. QQQ scans were acquired as MRM experiments with collision gas (nitrogen) set to medium. DP and CE for individual MRM transitions were carried out with further DP and CE optimization. A specific set of MRM transitions were monitored for each period according to the metabolites eluted within this period.

### 2.5. Phosphoproteomic Analysis

#### 2.5.1. Protein Extraction and Trypsin Digestion

The protein extraction and trypsin digestion were carried out as described previously [23,24]. Four volumes of buffer (1% Dephosphorylate Inhibitor, 10 mM dithiothreitol, and 1% Protease Inhibitor Cocktail) were added to the homogenized samples and treated by sonication three times. Then, an equal volume of tris-saturated phenol was added to the buffer for extraction and centrifuged at 5500× *g* for 10 min at 4 °C (RWD M1324, Shenzhen, China). The supernatant was precipitated overnight with five volumes of 0.1 M methanol. Finally, 8 M urea was used to dissolve the proteins and the BCA kit (Beyotime, Shanghai, China) was used to determine the concentration of proteins.

To induce protein digestion, the protein solution was reduced with 5 mM dithiothreitol at 56 °C for 30 min, and alkylated with 11 mM iodoacetamide in darkness for 15 min. Then, the samples were diluted to make sure that the urea concentration was less than 2 M. Trypsin was added at a mass ratio of 1:50 (trypsin: protein) for the first digestion overnight. Finally, trypsin was added in the mass ratio of 1:100 (trypsin: protein) for a second 4 h-digestion.

#### 2.5.2. Affinity Enrichment

The immobilized metal affinity chromatography (IMAC) microspheres were used to enrich phosphopeptides [25]. The peptides were dissolved in an enrichment buffer solution (50% acetonitrile/6% trifluoroacetic acid). The supernatant was transferred to the IMAC microspheres which were washed in advance, and then placed on a rotary shaker and shaken gently. Then, the IMAC microspheres were washed three times. Finally, the modified peptide was eluted with 10% ammonia water, and the eluent was collected and dried by vacuum freezing. After drying, the C18 ZipTips (Millipore, Burlington, MA, USA) was used to desalt.

#### 2.5.3. LC-MS/MS Analysis

The EASY-nLC 1000 UPLC system was used to separate after the peptides were dissolved in mobile phase A (0.1% (*v/v*) formic acid). Solvent A contained 0.1% formic acid and 2% acetonitrile, and Solvent B contained 0.1% formic acid and 90% acetonitrile. The gradient comprised an increase from 2% to 6% solvent B over 5 min, 6% to 18% in 40 min, 18% to 28% in 10 min, and climbed to 70% in 2 min, then held at 70% for 3 min, all at a constant flow rate of 0.4 μL/min.

The NSI source and Orbitrap Fusion Lumos mass spectrometry were used for the ionization and analysis of the peptides. The ion source voltage applied was 2.4 kV. The scanning range and scanning resolution of MS were set to 385–1500 m/z and 60,000. The scanning range and scanning resolution of MS/MS were set to 100 m/z and 15,000. Data-dependent procedure (DDA) was used for data collection. The automatic gain control (AGC) was set at 5E4 to improve the effective utilization of mass spectrometry. The signal threshold and maximum injection time were set to 15,000 ions/s and 30 ms.

#### 2.5.4. Database Search

The Maxquant search engine (v.1.5.2.8) was used to process the resulting MS/MS data. The database was *Camellia sinensis* database (33,932 sequences). Trypsin/P was specified as a cleavage enzyme allowing up to 2 missing cleavages. In the first search and main search, the mass error tolerance for precursor ions was 20 ppm and 5 ppm respectively, and that for fragment ions was set as 0.02 Da. The fixed modification was carbamidomethyl on Cys, the variable modifications were oxidation on Met, phosphorylation on Ser, Thr, and Tyr, and acetylation on the protein N-terminus and deamidation. The False discovery rate (FDR) was adjusted to 1%.

## 3. Results

### 3.1. Analysis of Phenotype and Physiological Indices of Tea Plants in Response to Glycine

To analyze the effect of glycine on the growth of tea leaves and roots, some physiological indices of tea roots and leaves were measured. As shown in Figure 1A, there were significant differences observed in tea roots. There was almost no new root formation in tea plants supplied with 1 mM Gly, while some new roots emerged in the control plants. It shows that Glycine used as the only nitrogen source, does not promote root growth of tea plants, and subsequently does not change root morphology. On the contrary, the color of tea leaves changed to a certain extent. The leaves of tea plants treated with Gly (GL) were greener than those in the control leaves treated without Gly (CL). Therefore, we measured the chlorophyll contents and net photosynthetic rate of tea plants, and the results are shown in Figure 1B. The contents of chlorophyll A, chlorophyll B, and total chlorophyll in GL were significantly higher than those in CL. Correspondingly, the net photosynthetic rate of leaves with Gly-supply was also significantly higher than that in leaves without Gly-supply. After all, the photosynthetic rate is directly proportional to the chlorophyll contents. In addition, to analyze the nitrogen nutritional function of Gly, the contents of total nitrogen were also detected in tea plants. As shown in Figure 1C, the contents of total nitrogen in tea roots with Gly-supply (GR) were significantly higher than that in tea roots without Gly-supply (CR). The same situation also occurred in leaves, and the total nitrogen content of GL was significantly higher than that of CL. Interestingly, there was no significant difference in nitrogen contents between CR and CL, while the nitrogen content of GR was significantly higher than that of GL. The results suggested that Gly as a nitrogen source could provide nitrogen nutrition for both tea roots and leaves, and Gly could also change the distribution of nitrogen between tea roots and leaves.

### 3.2. Glycine-Supply Increased Amino Acids Contents of Tea Plants

In order to analyze the effect of Gly-supply on amino acid metabolism in tea plants, we measured the contents of 19 amino acids in tea plants. In root samples, the content of each amino acid in GR was significantly higher than that in CR (Table 1), indicating that Gly-supply effectively improved amino acid metabolism. In leaves samples, the contents of methionine, isoleucine, arginine, leucine, phenylalanine, alanine, and asparagine in GL were significantly higher than those in CL, which was consistent with the changes in roots, indicating that Gly-supply also promoted the metabolic intensity of these amino acids. There was no significant difference in the contents of histidine, glutamic acid, threonine, proline, lysine, and glutamine between GL and CL, indicating that Gly-supply had little effect on the metabolic intensity of these amino acids of tea leaves. The contents of glycine, aspartic acid, tyrosine, valine, serine, and cysteine in GL were significantly lower than those in CL, indicating that Gly-supply might inhibit the metabolism of these amino acids in leaves. It should be noted that the contents of alanine, threonine, proline, glycine, aspartic acid, valine, serine, and cysteine in CL were significantly higher than those in CR, while the contents of these amino acids in GL were significantly lower than those in GR, indicating that Gly-supply changed the distribution of these amino acids between tea roots and leaves to a certain extent. In addition, we also focused on theanine and tryptophan in the metabolomic data. For theanine, there was no significant difference between GR and CR (Fold Change (GR/CR) was 0.94), while the intensity of GL was significantly higher than that of CL (Fold Change (GL/CL) was 9.52). For tryptophan, Gly-supply significantly reduced the intensity in tea roots and leaves.

### 3.3. Glycine-Supply Changed Enzyme Activities in Tea Plants

The activities of 12 enzymes were detected in tea plants (Table 2). The activity of GS in GL was significantly higher than in CL, while the activity of Fd-GOGAT in GL was significantly lower than in CL. The activity of SuSy-I (Decomposition) in GR was significantly lower than in CR, however, there was no significant difference between GL and CL. The activity of SuSy-II (Synthesis) in GR was slightly but not significantly higher than in CR, while the activity of SuSy-II in GL was significantly higher than in CL. In tea roots, the activities of GAPDH and NADO-ME in GR were significantly higher than in CR, while the activities of PGM and PEPC in GR were significantly lower than in CR. In tea leaves, TPS, HK, and PEPC had higher activities of enzymes in GL, while PGM, ALDO, GAPDH, PEPCK, and NADP-ME had higher activities of enzymes in CL.

### 3.4. Glycine-Supply Affected the Primary Metabolism of Tea Plants

Based on the UPLC-MS/MS analysis, a total of 521 metabolites were detected (Appendix A). We focused on 19 metabolites related to energy metabolism (Table 3). In the tea roots, although the intensities of sucrose, glucose, and fructose in GR were lower than in CR, the metabolites involved in glycolysis such as glucose 6-phosphate, glucose-1-phosphate, fructose-6-Phosphate, fructose-1,6-biphosphate, 3-phospho-glyceric acid and glyceraldehyde-3-phosphate were significantly higher than in CR. In addition, the intensities of metabolites involved in the TCA cycle such as malic acid, citric acid, isocitric acid, fumaric acid, and α-ketoglutaric acid in GR were lower than in CR. The intensities of metabolites in tea leaves were different from those in roots. Glycolysis-related metabolites such as glucose 6-phosphate, glucose-1-phosphate, fructose-6-phosphate, fructose-1,6-biphosphate, 3-phospho-glyceric acid, and glyceraldehyde-3-phosphate had lower intensities in GL, while TCA cycle-related metabolites such as citric acid, isocitric acid, and fumaric acid had higher intensities in GL. It was noteworthy that the intensity of α-ketoglutaric acid in GL was significantly lower than that in CL, which might be related to the direct involvement of α-ketoglutaric acid in amino acid metabolism.

### 3.5. Glycine-Supply Affected the Phosphoproteome of Tea Plants

To explore the response of phosphorylated protein to glycine, we analyzed the phosphoproteome in this study. A total of 3186 phosphorylation proteins and 8368 phosphorylation sites were identified. Among them, 4964 phosphorylation sites of 2555 phosphorylated proteins were quantified (Appendix A). We focused on some differential phosphorylation proteins and sites (Fold change > 1.50 or Fold change < 0.67, and *p*-value < 0.05) related to energy metabolism, amino acid metabolism, and photosynthesis (Table 4). The phosphorylation levels of energy metabolism-related proteins in GR were significantly lower than in CR, indicating that Gly-supply regulated energy metabolism in tea roots through dephosphorylation. The phosphorylation sites of three photosynthesis-related proteins, photosystem II D1 protein, chlorophyll a-b binding protein CP29.1, and ferredoxin-NADP reductase, were up-regulated in GL, indicating that Gly-supply had a certain effect on the photosynthesis of tea leaves through protein phosphorylation. The phosphorylation of GLDC on Thr-98 was down-regulated in GL, while the phosphorylation of AMT on Ser-173 was up-regulated. In addition, two phosphorylation sites, Thr-99 of glutamine synthetase (GS) and Ser-1100 of ferredoxin-dependent glutamate synthase (Fd-GOGAT) were significantly up-regulated in GL. These results suggested that Gly-supply might have affected glycine cleavage and ammonia assimilation in tea leaves by changing the phosphorylation levels of these proteins.

## 4. Discussion

### 4.1. Glycine-Supply Regulated the Formation of New Roots and Promoted the Photosynthesis of Tea Leaves

There are various effects of nitrogen form on root development. Based on our study, the contents of amino acids and total nitrogen in GR were significantly increased, indicating that tea roots did absorb Gly which could supply nitrogen nutrition for tea plants. However, there was no new root formation in tea plants supplied with 1 mM Gly. Therefore, we speculated that Gly could be used not only as a nitrogen source for the growth of tea plants but also as an important signal molecule to regulate the growth of tea plants in our study. A study by Han showed that exogenous glycine (2.5 mM) inhibited the root elongation of pak choi in hydroponic culture [11]. In another study by Fedoreyeva, Gly in a medium with a concentration of 10^−4^ mM significantly stimulate the growth and development of tobacco calli, however, with the increase of Gly concentration to 10^−3^–10^−2^ mM, the yield of the tobacco callus fresh weight and the number of regenerants per explant decreased [26]. These results showed that if the concentration of glycine or amino acid is not proper, it will have adverse effects on the growth of plant roots. Generally, the concentrations of amino acids are very low in soil solutions. However, in soils that absorb substantial quantities of amino acids and decompose organic matter (organic fertilizer), the concentrations of amino acids may exceed the levels required for the normal growth and development of plant roots [27]. This is a problem that needs special attention in actual production.

In general, chlorophyll content is positively correlated with photosynthetic intensity, and the stronger the photosynthesis, the more vigorous the growth of plant leaves. Based on the results of the physiological indexes above, GL had higher chlorophyll contents and a stronger photosynthetic rate, which meant that tea leaves with Gly-supply had better growth.

In conclusion, 1 mM Gly might inhibit the formation of new roots of tea plants but promote the growth of tea leaves. Therefore, exploring the regulation mechanism of glycine on the growth of tea roots and leaves could provide a scientific basis and guidance for the combined application of inorganic fertilizer and organic fertilizer in organic agriculture.

### 4.2. Glycine-Supply Affected the Energy Metabolism by Phosphorylation of Proteins in Tea Roots

#### 4.2.1. Gly-Supply Promoted the Accumulation of Glc6P in Sucrose Metabolism

The cleavage of sucrose (Suc) is catalyzed either by invertases or by sucrose synthases. In this study, sucrose synthase 2 (TEA017533.1, SuSy, EC 2.4.1.13) was identified and its phosphorylation level was significantly different between GR and CR. SuSy catalyzed the reversible cleavage of Suc using UDP to yield fructose (Fru) and UDP-glucose (UDP-Glc) (Figure 2). The SuSy was reported to contain two phosphorylation sites. The first site is a serine phosphorylation site at positions 11 to 15, which is thought to play a role in protein solubility and enzyme activity. The second site is also a serine, at around position 170, and is thought to regulate protein degradation [28,29]. In our study, two phosphorylation sites of SuSy, Ser-11, and Ser-151, were detected, and both two phosphorylation sites were significantly down-regulated in GR. The dephosphorylation of Ser-11 might cause SuSy to be less soluble [29] and reduce the sucrose cleavage activity of SuSy [30,31]. Indeed, the determination of sucrose synthase activity also showed that the activity of SuSy in the decomposition direction was decreased significantly in GR. Ser-151 might be the second phosphorylation site specific to tea plants and the dephosphorylation of Ser-151 might cause the degradation of SuSy [28,29,30]. In brief, Gly-supply could decrease the phosphorylation level of SuSy, thus reducing the decomposition activity of SuSy in GR.

Trehalose-phosphate synthase (TPS, EC 2.4.1.15) can catalyze the synthesis of trehalose-6-phosphate (Tre6P) from UDP-Glc and glucose-6-phosphate (Glc6P). Tre6P potently inhibits the activity of hexokinase (HK). Without Tre6P, ATP is expended faster by HK than it can be replenished by later steps of glycolysis [32]. In this study, TPS (TEA006786.1, alpha, alpha-trehalose-phosphate synthase [UDP-forming] 9 isoform X1) was phosphorylated at Ser-5 and down-regulated in GR. Little research has been conducted on the phosphorylation of TPS. However, other studies have shown that phosphorylation can lead to reduced activity of other phosphate synthases, such as sucrose-phosphate synthase [33] and myo-Inositol-3-phosphate synthase [34]. So, we guessed that the down-regulated Ser-5 in GR might enhance the activity of TPS and promote the synthesis of Tre6P. The enzyme activity of TPS in GR was slightly higher than that in CR, and GR did have a higher Tre6P accumulation.

Phosphoglucomutase (PGM, EC 5.4.2.2) catalyzes the interconversion of glucose-1-phosphate (Glc1P) and glucose-6-phosphate (Glc6P). Studies have shown that the post-translational phosphorylation of PGM can enhance its activity, and PGM preferentially converts Glc1P to Glc6P [35,36,37]. In this study, PGM (TEA018596.1) was modified by phosphorylation on Ser-193, and the phosphorylation level of the site was down-regulated in GR, and the results of enzyme activity assay showed that the activity of PGM in GR was indeed decreased.

In terms of metabolism, the intensity of sucrose in GR was low, but the intensity of Glc1P and Glc6P was high. This might be because other metabolic pathways in GR were conducive to the accumulation of substances such as Glc1P and Glc6P. It should be noted that Glc6P is the direct substrate of glycolysis. Therefore, the higher Glc6P accumulation in GR provided a sufficient material basis for the glycolysis pathway.

#### 4.2.2. Gly-Supply Increased the Activity of Glycolysis

Glycolysis is a sequence of enzymatic reactions. Most glycolytic enzymes have been reported to be phosphorylated and phosphorylation increases enzymes’ activity or stability [38]. In this study, fructose-bisphosphate aldolase cytoplasmic isozyme (TEA023229.1, ALDO, and EC 4.1.2.13), glyceraldehyde-3-phosphate dehydrogenase (TEA031641.1, GAPDH, and EC 1.2.1.12), and 2,3-bisphosphoglycerate-independent phosphoglycerate mutase (TEA013943.1, iPGAM, and EC 5.4.2.12) were identified by phosphoproteome.

ALDO contained four phosphorylation sites, Ser-76, Ser-85, Ser-384, and Ser-394, while iPGAM contained two phosphorylation sites, Ser-3 and Ser-69. All phosphorylation levels of these sites were significantly downregulated in GR. The dephosphorylation was reported to reduce the enzyme activity of ALDO and iPGAM [37,38,39,40]. However, in this study, there was no significant difference in the enzyme activity of ALDO between GR and CR. The report had shown that dephosphorylation of GAPDH can enhance its enzyme activity [41]. In our study, GAPDH contained one phosphorylation site, Ser-306, and its phosphorylation level was significantly down-regulated in GR. Therefore, it was speculated that the activity of GAPDH was increased in GR. Indeed, the determination result of GAPDH enzyme activity verified this speculation. GAPDH can catalyze the conversion of glyceraldehyde-3-phosphate (Ga3P) to 1,3-bisphosphate-glycerate (1,3PGA). Then, the 1,3PGA was metabolized by phosphoglycerate kinase (PGK, EC 2.7.2.3) to produce ATP and 3-phosphoglycerate (3PGA). Interestingly, Ga3P could also be directly oxidized to 3PGA by the non-phosphorylating glyceraldehyde-3-phosphate dehydrogenase (np-GAPDH, EC 1.2.1.9) (Figure 2). These two pathways occurred simultaneously, in order to effectively regulate the cellular production of energy (ATP and NADH) and reductive (NADPH) power according to transitory cell requirements [41,42]. Therefore, we speculated that Gly-supply induced dephosphorylation of GAPDH might be to synthesize as much ATP as possible.

In addition to the difference in enzyme activity, we also noted that the intensity of metabolites in the glycolysis pathway such as Fru1,6P, Ga3P, and 3PGA were significantly higher in GR than in CR. These results indicated that glycolysis was more active in GR than that in CR.

#### 4.2.3. Gly-Supply Regulated Pyruvate Metabolism and Decreased the Activity of TCA Cycle

In this study, four enzymes related to pyruvate metabolism were detected by phosphoproteome, which showed a significant difference in GR compared with CR. They were phosphoenolpyruvate carboxylase (TEA009852.1, PEPC, EC 4.1.1.31, Ser-11, and Ser-966), phosphoenolpyruvate carboxykinase (TEA008970.1, PEPCK, EC 4.1.1.49, and Ser-57), NADP-dependent malic enzyme (TEA003598.1, NADP-ME, EC 1.1.1.40, and Ser-357) and acetyl-coenzyme A synthetase (TEA005053.1, ACSS, EC 6.2.1.1, and Ser-125). Phosphorylation levels of corresponding sites of the four enzymes were down-regulated in GR.

Phosphoenolpyruvate (PEP) is a high-energy metabolic intermediate and is important to connect a variety of metabolic pathways, such as glycolysis, gluconeogenesis, and organic acid metabolism. PEPC and PEPCK are directly related to PEP accumulation. PEPC catalyzes the irreversible *β*-carboxylation of phosphoenolpyruvate (PEP) in the presence of HCO_3^−^_ to yield oxaloacetate (OAA) and inorganic phosphate (Pi), while PEPCK catalyzes the reversible decarboxylation of OAA to yield PEP and CO_2_. Here, the dephosphorylation of PEPC might result in an increased sensitivity of PEPC to the inhibition of malate (Mal) and thus decrease the activity of PEPC, as a previous study reported [43]. The results of enzyme activity showed that PEPC activity decreased significantly in GR. By contrast, dephosphorylation increased PEPCK activity and promoted the synthesis of PEP [44]. The intensity of PEP was indeed higher in GR. So, we speculated that the dephosphorylation of PEPC and PEPCK was a way to regulate the opposite reactions between PEPC and PEPCK and avoid futile cycles, and promoted the reaction to the direction of PEP synthesis in GR.

Pyruvate (PA) is the key to linking carbohydrate metabolism, fatty acid metabolism, and amino acid metabolism. NADP-ME catalyzes the oxidative decarboxylation of Mal using NADP^+^ as a coenzyme in the presence of divalent metal ions to produce PA, NADPH, and CO_2_. The enzyme activity of NADP-ME in GR was significantly higher than that in CR because the dephosphorylation increased the affinity between NADP-ME and NADP^+^ [45,46]. This regulation was beneficial to the synthesis of PA. As PA was directly related to the synthesis of valine, leucine, and isoleucine, the regulation promoted the accumulation of these amino acids, which was consistent with the vigorous metabolism of amino acids in GR. ACSS is one of the enzymes that catalyze the production of acetyl-CoA, and the dephosphorylation would reduce the activity of ACSS [47,48]. So, the decreased phosphorylation of ACSS might lead to the decrease of acetyl-CoA synthesis in GR, which eventually led to the decrease of TCA cycle intensity, because acetyl-CoA was the beginning of the TCA cycle. Interestingly, the intensities of citric acid, isocitric acid, α-ketoglutaric acid, and fumaric acid were lower in GR than those in CR, which meant that the activity of the TCA cycle in GR was indeed lower than that in CR.

In conclusion, Gly-supply regulated the energy metabolism of tea roots mainly from two aspects. Firstly, Gly-supply induced the dephosphorylation of ALDO and GAPDH, which enhanced their enzyme activities, and led to more active glycolysis. Secondly, Gly-supply regulated the dephosphorylation of PEPC, PEPCK, NADP-ME, and ACSS, which was conducive to the synthesis of PEP and PA, but not conducive to the accumulation of acetyl-CoA, resulting in enhanced nitrogen metabolism and decreased TCA cycle intensity. In plants, both glycolysis and the TCA cycle can provide energy, but the TCA cycle is the metabolic hub of sugars, lipids, and proteins, which plays a key role in providing substrates for metabolite synthesis or signals for feedback. Therefore, the decrease of TCA cycle intensity in GR might lead to the inactivity in lipids and protein syntheses, resulting in the obstruction of root growth, while the energy generated by enhanced glycolysis might be mainly used for the upward transportation of substances such as amino acids.

### 4.3. Glycine-Supply Promoted Photosynthesis in Tea Leaves and Regulated Amino Acid Metabolism to Improve Tea Quality

#### 4.3.1. Gly-Supply Promoted the Photosynthesis

Photosystem II (PSII) is a key pigment-protein complex involved in light reactions of photosynthesis. One of the regulatory mechanisms of PSII is the reversible phosphorylation of PSII core protein [49]. The D1 protein is the main member of the PSII core, and its phosphorylation is a prerequisite for the efficient migration of damaged PSII complexes from grana to stroma lamellae for repair [50]. The damaged D1 protein is then degraded by D1 proteases and removed from PSII. Finally, a new D1 protein is reassembled into PSII [51,52]. In our study, the D1 protein (TEA001596.1) had two phosphorylation sites significantly up-regulated in GL, Thr-2, and Ser-232. These results indicated that the Gly-organic nitrogen could avoid the net loss of D1 protein and improve the stability and rapid repair of PSII by increasing the phosphorylation level of D1 protein in tea leaves.

In addition to the D1 protein, the phosphorylation level of CP29 (TEA032678.1, Thr-108), an antenna protein of PSII, was also significantly up-regulated in GL. The phosphorylation of CP29 could enhance non-photochemical quenching (NPQ) [53,54]. Studies have shown that the process of NPQ is rather aimed at photoprotection from excess light [55]. So, the up-regulated phosphorylation of CP29 had a protective effect on photosynthesis by enhancing the excess energy dissipation.

Ferredoxin-NADP^+^ reductase catalyzes the last step of photosynthetic electron transport in chloroplasts, driving electrons from reduced ferredoxin to NADP^+^ [56]. In this study, the phosphorylation site of ferredoxin-NADP^+^ reductase (TEA032600.1, FNR, EC 1.18.1.2), Thr-171, was up-regulated in GL. This regulation could improve photosynthesis because it enhanced the electron transfer ability of FNR [57].

In general, changes in the phosphorylation levels of three photosynthesis-related proteins, D1 protein, CP29, and FNR, promoted photosynthesis in GL to a certain extent. These results are consistent with the photosynthetic rate measured in this experiment.

#### 4.3.2. Gly-Supply Promoted the Cleavage and Reuse of Glycine

Glycine catabolism is mediated primarily by the glycine cleavage system, which comprises four proteins: glycine dehydrogenase (decarboxylating) (GLDC), aminomethyltransferase (AMT), dihydrolipoyl dehydrogenase (DLD), and glycine cleavage system protein H (GCSH) [58]. GLDC, AMT, and DLD decompose a molecule of glycine into a molecule of carbon dioxide, ammonia, and a carbon unit with the aid of carrier protein GCSH. In our research, compared with CL, the phosphorylation levels of GLDC (TEA022478.1, EC 1.4.4.2) and AMT (TEA023186.1, EC 2.1.2.10) in GL were significantly different. The phosphorylation of GLDC on Thr-98 was down-regulated in GL, while the phosphorylation of AMT on Ser-173 was up-regulated. Although few studies have been conducted on the phosphorylation of GLDC and AMT, the regulation of phosphorylation of enzymes with similar functions may be similar. GAPDH, mentioned above, is similar to GLDC in that they are both dehydrogenases. The phosphorylation of GAPDH could reduce the enzyme activity [41], so the down-regulation of GLDC phosphorylation on Thr-98 might enhance the enzyme activity of GLDC. Similarly, since phosphorylation could enhance the enzyme activity of other transferases, such as glutathione *S*-transferase [59], the enzyme activity of AMT might be enhanced after phosphorylation modification. In brief, the Gly-supply might enhance the enzyme activities of GLDC and AMT, accelerate the catabolism of glycine, and promote the metabolism of amino acids in tea leaves.

In plants, ammonia assimilation takes place via the glutamine synthetase/glutamate synthase cycle (GS/GOGAT). The free ammonia combines with glutamate to form glutamine by glutamine synthetase at the expense of ATP. Then, an amido group from glutamine is transferred to 2-oxoglutarate to form two molecules of glutamate under the catalysis of glutamate synthase. Studies have shown that the phosphorylation of GS/GOGAT enzymes may constitute a regulatory mechanism controlling the activity of this ammonia assimilation cycle [60]. In our study, two phosphorylation sites, Thr-99 of glutamine synthetase (TEA028194.1, GS, EC 6.3.1.2) and Ser-1100 of ferredoxin-dependent glutamate synthase (TEA030315.1, Fd-GOGAT, EC 1.4.7.1), were significantly up-regulated in GL. Some studies had shown that the phosphorylation of GS on Ser might lead to the reduction of enzyme activity [61,62]. However, in this study, GS was phosphorylated on Thr, and the enzyme activity of GS in GL was significantly higher than that in CL (Table 2). There are few studies about the regulation of Fd-GOGAT by phosphorylation. However, the determination of the enzyme activity of Fd-GOGAT indicated that phosphorylation might decrease its enzyme activity. These results indicated that the tea plants with Gly-supply could enhance the ability of ammonia assimilation and improve the utilization rate of inorganic nitrogen.

In summary, the utilization of Gly-organic nitrogen in tea plants might be divided into two steps. First, tea plants improved the enzyme activities of GLDC and AMT by dephosphorylation and phosphorylation, respectively, accelerated the decomposition of glycine, and obtained a large amount of free ammonia. Then, the enzyme activity of GS was increased by phosphorylation, and the ability of ammonia assimilation was enhanced so that free ammonia could be transformed into organic nitrogen (glutamine). The decrease of Fd-GOGAT enzyme activity caused by phosphorylation might be to avoid the ineffective cycle between glutamine and glutamate and promote glutamine to participate in other metabolic activities, such as amino sugar and nucleotide sugar metabolism, purine metabolism, and so on.

#### 4.3.3. Gly-Supply Improved Tea Quality

Many studies have shown that amino acids play important roles in the characteristic flavor and subtle taste of tea [22,63,64,65]. Theanine, the most abundant and important amino acid in tea, exists only in its free form and accounts for 50% of the total amino acid content. Theanine can reduce the bitter and astringent taste of tea infusion and improve the sweet and umami taste [66,67,68]. So, the content of theanine is significantly positively correlated with tea quality [69,70]. In this study, the intensity of theanine was significantly higher in GL than that in CL (Table 1, Fold Change (GL/CL) was 9.52). This might be because of the enhanced activity of GS in GL, and GS had been reported to be one of the key enzymes in the synthesis of theanine in a recent study [71]. In addition, glutamic acid also has an umami taste, which can also improve the taste of tea infusions [72]. Its content in GL was slightly higher than that in CL. Alanine has a sweet taste [73], and its content in GL was significantly higher than that in CL. Valine and tryptophan have a bitter taste [73], and their intensity in GL was significantly lower than that in CL. For brevity, Gly-supply regulated amino acid metabolism to highlight the sweet and umami taste of theanine, glutamate, and alanine, and reduce the bitter taste of valine and tryptophan. This had a positive effect on improving the quality of tea.

## 5. Conclusions

In this study, we compared the differences in biochemical composition, enzyme activity, metabolome, and phosphoproteome between the experimental group and the control group, and finally reached the following important conclusions: (i) In roots, glycine-supply induced dephosphorylation of proteins, such as fructose-bisphosphate aldolase cytoplasmic isozyme, glyceraldehyde-3-phosphate dehydrogenase, and phosphoenolpyruvate carboxylase, resulting in increased intensity of glycolysis and decreased intensity of TCA cycle. (ii) In leaves, glycine-supply changed the phosphorylation levels of glycine dehydrogenase, aminomethyltransferase, glutamine synthetase, and ferredoxin-dependent glutamate synthase, which accelerated the decomposition of glycine and enhanced the ability of ammonia assimilation. (iii) Glycine- supply could improve the tea quality by increasing the intensity of amino acids, such as theanine and alanine. This study not only clarified the modification mechanism of protein phosphorylation in the process of glycine absorption by tea plants but also clarified the important regulatory mechanism of amino acid nitrogen on the growth and development of tea plants, establishing an important theoretical foundation for the utilization of amino acid nitrogen by tea plants.

## Figures and Tables

**Figure 1 foods-12-00334-f001:**
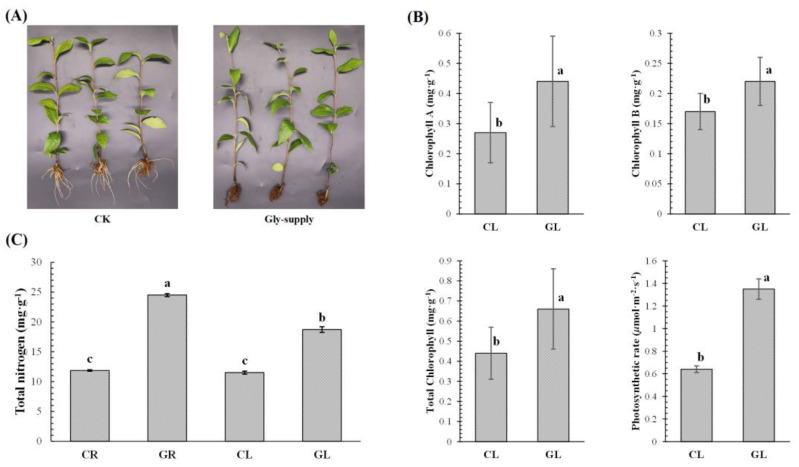
(**A**) Phenotypes of tea plants. (**B**) The chlorophyll contents and net photosynthetic rates of tea leaves. (**C**) The nitrogen contents of tea roots and leaves. Different lowercase letters indicated significant differences according to one-way ANOVA (*p* < 0.05).

**Figure 2 foods-12-00334-f002:**
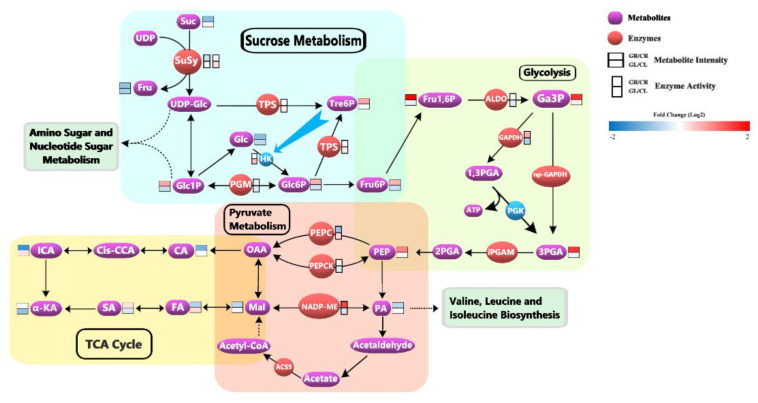
Different phosphoproteins and metabolites are involved in the metabolism of tea plants. Suc: sucrose; SuSy: sucrose synthase 2; Fru: fructose; UDP–Glc: UDP–glucose; TPS: trehalose–phosphate synthase; PGM: phosphoglucomutase; Tre6P: trehalose–6–phosphate; Glc1P: glucose–1–phosphate; Glc: glucose; HK: hexokinase; Glc6P: glucose–6–phosphate; Fru6P: fructose–6–phosphate; Fru1,6P: fructose–1, 6–phosphate; ALDO: fructose–bisphosphate aldolase cytoplasmic isozyme; Ga3P: glyceraldehyde–3–phosphate;GAPDH: glyceraldehyde–3–phosphate dehydrogenase; np–GAPDH: non-phosphorylating glyceraldehyde–3–phosphate dehydrogenase; 1,3PGA: 1,3–bisphosph–glycerate; PGK: phosphoglycerate kinase; 3PGA: 3–phosphop–glycerate; iPGAM: 2,3–bisphosphoglycerate–independent phosphoglycerate mutase; 2PGA: 2–phosphop−glycerate; PEP: phosphoenolpyruvate; OAA: oxaloacetate; PEPC: phosphoenolpyruvate carboxylase; PEPCK: phosphoenolpyruvate carboxykinase; PA: pyruvic acid; Mal: malate; NADP–ME: NADP–dependent malic enzyme; ACSS: acetyl–coenzyme A synthetase; CA: citric acid; Cis–CCA: cis–aconitate; ICA: isocitric acid; α–KA: α–ketoglutaric acid; SA: succinic acid; FA: fumaric acid.

**Table 1 foods-12-00334-t001:** Contents of amino acids in tea roots and leaves.

	GR	CR	GL	CL
Methionine (μg/g)	338.38 ± 3.49 ^a^	87.19 ± 2.78 ^c^	328.48 ± 8.59 ^a^	164.54 ± 2.44 ^b^
Isoleucine (μg/g)	115.48 ± 1.08 ^c^	30.06 ± 0.61 ^d^	163.87 ± 3.80 ^a^	123.86 ± 1.73 ^b^
Arginine (μg/g)	200.62 ± 2.12 ^b^	46.13 ± 1.61 ^c^	263.84 ± 14.70 ^a^	216.54 ± 3.37 ^b^
Leucine (μg/g)	3.50 ± 0.02 ^c^	2.58 ± 0.04 ^d^	5.47 ± 0.48 ^a^	4.82 ± 0.05 ^b^
Phenylalanine (μg/g)	395.38 ± 3.93 ^c^	95.61 ± 2.56 ^d^	494.10 ± 10.90 ^a^	449.26 ± 6.65 ^b^
Alanine (μg/g)	357.51 ± 3.65 ^a^	68.17 ± 1.99 ^d^	331.77 ± 6.88 ^b^	319.13 ± 4.79 ^c^
Asparagine (μg/g)	371.46 ± 1.92 ^c^	226.62 ± 1.31 ^d^	434.48 ± 5.09 ^a^	417.73 ± 3.56 ^b^
Histidine (μg/g)	1216.24 ± 12.72 ^b^	234.87 ± 7.98 ^c^	1315.88 ± 20.14 ^a^	1308.85 ± 20.28 ^a^
Glutamic acid (μg/g)	1218.66 ± 12.81 ^a^	237.73 ± 8.15 ^c^	1199.58 ± 16.91 ^ab^	1183.43 ± 18.36 ^b^
Threonine (μg/g)	365.46 ± 3.68 ^a^	65.22 ± 1.72 ^c^	348.32 ± 4.80 ^b^	346.86 ± 5.15 ^b^
Proline (μg/g)	816.44 ± 8.61 ^a^	141.24 ± 4.92 ^c^	775.21 ± 6.79 ^b^	782.46 ± 12.18 ^b^
Lysine (μg/g)	486.39 ± 4.96 ^b^	118.44 ± 3.58 ^c^	626.35 ± 7.81 ^a^	632.34 ± 9.60 ^a^
Glutamine (μg/g)	1080.90 ± 9.72 ^b^	338.72 ± 6.29 ^c^	1177.78 ± 18.21 ^a^	1186.80 ± 16.01 ^a^
Glycine (μg/g)	496.69 ± 5.19 ^a^	108.87 ± 3.65 ^d^	352.13 ± 12.28 ^c^	383.01 ± 5.89 ^b^
Aspartic acid (μg/g)	993.61 ± 10.22 ^a^	201.08 ± 6.14 ^d^	631.53 ± 12.92 ^c^	749.93 ± 11.28 ^b^
Tyrosine (μg/g)	807.84 ± 8.41 ^c^	170.90 ± 5.59 ^d^	975.01 ± 28.25 ^b^	1160.41 ± 17.90 ^a^
Valine (μg/g)	926.92 ± 9.64 ^a^	179.49 ± 5.80 ^d^	454.69 ± 8.32 ^c^	549.42 ± 8.34 ^b^
Serine (μg/g)	38.84 ± 0.31 ^a^	15.22 ± 0.20 ^b^	12.2 ± 0.22 ^c^	15.62 ± 0.10 ^b^
Cysteine (μg/g)	1188.72 ± 11.06 ^a^	661.33 ± 18.28 ^c^	582.93 ± 9.83 ^d^	1087.27 ± 14.75 ^b^
Theanine ^#^	1,805,733 ± 86,848.00 ^b^	1,922,700 ± 57,051.26 ^b^	1,125,000 ± 19,214.75 ^a^	118,200 ± 3167.09 ^c^
Tryptophan ^#^	10,435,000 ± 332,349.21 ^b^	36,827,333 ± 532,304.63 ^a^	1,655,700 ± 23,693.46 ^d^	4,064,633 ± 35,089.44 ^c^

Contents of amino acids were presented as mean ± standard deviation. Different lowercase letters indicated significant differences according to the Duncan test at *p* < 0.05. **^#^** The data came from the results of metabolomics (relative value).

**Table 2 foods-12-00334-t002:** The activities of enzymes in tea plants.

Enzymes	GR	CR	GL	CL
GS (U·g^−1^)	-	-	58.61 ± 8.85 ^a^	24.69 ± 0.41 ^b^
Fd-GOGAT (nmol·min^−1^·g^−1^)	-	-	21.54 ± 1.18 ^b^	36.86 ± 1.06 ^a^
SuSy-I (Decomposition, μg·min^−1^·g^−1^)	120.49 ± 3.59 ^c^	158.06 ± 2.90 ^b^	470.96 ± 7.90 ^a^	462.32 ± 10.69 ^a^
SuSy-II (Synthesis, μg·min^−1^·g^−1^)	490.06 ± 23.53 ^c^	470.73 ± 27.63 ^c^	1117.81 ± 50.36 ^a^	959.36 ± 46.82 ^b^
TPS (nmol·min^−1^·g^−1^)	23.04 ± 0.64 ^a^	22.50 ± 0.95 ^a^	18.02 ± 0.23 ^b^	16.31 ± 0.96 ^c^
PGM (nmol·min^−1^·g^−1^)	22.28 ± 0.36 ^d^	25.08 ± 0.67 ^c^	84.40 ± 2.27 ^b^	102.67 ± 2.03 ^a^
HK (nmol·min^−1^·g^−1^)	192.04 ± 3.40 ^a^	191.64 ± 9.84 ^a^	168.49 ± 10.56 ^b^	118.45 ± 3.91 ^c^
ALDO (nmol·min^−1^·g^−1^)	45.67 ± 3.12 ^c^	42.56 ± 0.84 ^c^	206.36 ± 0.94 ^b^	239.06 ± 14.30 ^a^
GAPDH (nmol·min^−1^·g^−1^)	34.89 ± 1.52 ^b^	22.59 ± 1.53 ^c^	33.81 ± 1.96 ^b^	51.21 ± 3.86 ^a^
PEPC (nmol·min^−1^·g^−1^)	21.17 ± 1.53 ^d^	36.34 ± 2.47 ^c^	728.21 ± 11.84 ^a^	595.28 ± 12.77 ^b^
PEPCK (nmol·min^−1^·g^−1^)	831.21 ± 37.99 ^a^	791.33 ± 60.96 ^a^	448.56 ± 25.60 ^c^	535.70 ± 17.79 ^b^
NADP-ME (nmol·min^−1^·g^−1^)	80.15 ± 4.19 ^c^	29.12 ± 1.46 ^d^	105.25 ± 6.14 ^b^	128.40 ± 7.74 ^a^

The activities of enzymes were presented as mean ± standard deviation. Different lowercase letters indicated significant differences according to the Duncan test at *p* < 0.05.

**Table 3 foods-12-00334-t003:** Differential metabolites associated with energy metabolism.

Compounds	Formula	Fold Change (GR/CR)	Fold Change (GL/CL)
D-Sucrose	C_12_H_22_O_11_	0.53	1.08
D-Glucose	C_6_H_12_O_6_	0.54	0.72
D-Fructose	C_6_H_12_O_6_	0.57	0.63
D-Trehalose	C_12_H_22_O_11_	0.64	1.32
Trehalose-6-phosphate	C_12_H_23_O_14_P	1.60	0.99
D-Glucose-6-phosphate	C_6_H_13_O_9_P	1.51	0.76
Glucose-1-phosphate	C_6_H_13_O_9_P	1.61	0.73
D-Fructose-6-phosphate	C_6_H_13_O_9_P	1.48	0.75
D-Fructose-1,6-biphosphate	C_6_H_14_O_12_P_2_	4.18	0.88
3-Phospho-D-glyceric acid	C_3_H_7_O_7_P	2.93	0.93
DL-Glyceraldehyde-3-phosphate	C_3_H_7_O_6_P	3.10	1.10
Pyruvic acid	C_3_H_4_O_3_	0.69	1.04
Phosphoenolpyruvate	C_3_H_5_O_6_P	1.95	1.05
D-Malic acid	C_4_H_6_O_5_	0.71	0.95
Citric acid	C_6_H_8_O_7_	0.48	1.02
Isocitric acid	C_6_H_8_O_7_	0.35	1.18
Fumaric acid	C_4_H_4_O_4_	0.67	1.23
Succinic acid	C_4_H_6_O_4_	1.16	0.82
α-Ketoglutaric acid	C_5_H_6_O_5_	0.97	0.56

**Table 4 foods-12-00334-t004:** Differential phosphorylation proteins and sites related to energy metabolism, amino acid metabolism, and photosynthesis.

Protein Accession	Phosphorylation Protein	Site	FC (GR/CR)	FC (GL/CL)
TEA017533.1	Sucrose synthase 2 (SuSy)	Ser-11 Ser-151	0.262 0.535	- -
TEA006786.1	Alpha, alpha-trehalose-phosphate synthase (TPS)	Ser-5	0.451	-
TEA018596.1	Phosphoglucomutase (PGM)	Ser-193	0.478	-
TEA023229.1	Fructose-bisphosphate aldolase cytoplasmic isozyme (ALDO)	Ser-76 Ser-85 Ser-384 Ser-394	0.613 0.443 0.563 0.340	0.723 - - -
TEA031641.1	Glyceraldehyde-3-phosphate dehydrogenase (GAPDH)	Ser-306	0.660	-
TEA013943.1	2,3-bisphosphoglycerate-independent phosphoglycerate mutase (iPGAM)	Ser-3 Ser-4 Ser-69	0.509 0.444 0.449	- - -
TEA009852.1	Phosphoenolpyruvate carboxylase (PEPC)	Ser-11 Ser-966	0.250 0.165	- -
TEA008970.1	Phosphoenolpyruvate carboxykinase (PEPCK)	Ser-57	0.454	-
TEA003598.1	NADP-dependent malic enzyme (NADP-ME)	Ser-357	0.538	-
TEA005053.1	Acetyl-coenzyme A synthetase (ACSS)	Ser-125	0.652	-
TEA001596.1	Photosystem II D1 protein	Thr-2 Ser-232	- -	2.478 4.961
TEA032678.1	Chlorophyll a-b binding protein CP29.1	Thr-108	-	2.948
TEA032600.1	Ferredoxin-NADP reductase (FNR)	Thr-171	-	3.268
TEA022478.1	Glycine dehydrogenase (decarboxylating) (GLDC)	Thr-98	-	0.369
TEA023186.1	Aminomethyltransferase (AMT)	Ser-173	-	1.668
TEA028194.1	Glutamine synthetase (GS)	Thr-99	-	2.797
TEA030315.1	Ferredoxin-dependent glutamate synthase (Fd-GOGAT)	Ser-1100	-	1.808

## Data Availability

The data are contained within the article. Other data used to support the findings of this study can be made available by the corresponding author upon request.

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
