# Peer review of "Glycine-Induced Phosphorylation Plays a Pivotal Role in Energy Metabolism in Roots and Amino Acid Metabolism in Leaves of Tea Plant"

_foods, 2023, doi:10.3390/foods12020334_

Round 1

Reviewer 1 Report

This is an interesting manuscript, because study the regulatory mechanisms of phosphorylation.  However, there isn’t an experimental design and the statistical analyses weren’t described rightly. Doubts and comments are in manuscript.

Best regards

Reviewer 2 Report

The paper titled “Glycine-induced phosphorylation plays a pivotal role in energy metabolism in roots and amino acid metabolism in leaves of tea plant”, represents a valuable, interesting, and well-written work. in my opinion the article can be considered for publication in this journal. Only some changes are required to the authors.

Abstract, line 23: write the acronym TCA cycle in full

Pag.4 line 147: 100 mm); The mobile phase…. Check punctuation

Pag.3, line 121 and Pag.4 line 175: add make and model of centrifuge

Pag.5 line 200, 400 nL/min????

Insert space between the different paragraphs.

I advise the authors to include the conclusions in a separate paragraph.

References: Check the format of some bibliographical references.

1.        Author 1, A.B.; Author 2, C.D. Title of the article. Abbreviated Journal Name Year, Volume, page range.
